# The Chemoautotrophically Based Movile Cave Groundwater Ecosystem, a Hotspot of Subterranean Biodiversity

**Traian Brad** [1,2,†] , **Sanda Iepure** [1,2,†] and **Serban M. Sarbu** [3,4,*,†]

1 "Emil Racoviță" Institute of Speleology, str. Clinicilor nr. 5-7, 400006 Cluj-Napoca, Romania; traian.brad@academia-cj.ro (T.B.); sanda.iepure@academia-cj.ro (S.I.)
2 Institutul Român de Știință și Tehnologie, str. Virgil Fulicea nr. 3, 400022 Cluj-Napoca, Romania
3 "Emil Racoviță" Institute of Speleology, str. Frumoasă nr. 31, 010986 Bucureşti, Romania
4 Department of Biological Sciences, California State University, Chico, CA 95929, USA
* Correspondence: serban.sarbu@yahoo.com
† All the three authors have the same contribution.

**Abstract:** Movile Cave hosts one of the world's most diverse subsurface invertebrate communities. In the absence of matter and energy input from the surface, this ecosystem relies entirely on in situ primary productivity by chemoautotrophic microorganisms. The energy source for these microorganisms is the oxidation of hydrogen sulfide provided continuously from the deep thermomineral aquifer, alongside methane, and ammonium. The microbial biofilms that cover the water surface, the cave walls, and the sediments, along with the free-swimming microorganisms, represent the food that protists, rotifers, nematodes, gastropods, and crustacean rely on. Voracious water-scorpions, leeches, and planarians form the peak of the aquatic food web in Movile Cave. The terrestrial community is even more diverse. It is composed of various species of worms, isopods, pseudoscorpions, spiders, centipedes, millipedes, springtails, diplurans, and beetles. An updated list of invertebrate species thriving in Movile Cave is provided herein. With 52 invertebrate species (21 aquatic and 31 terrestrial), of which 37 are endemic for this unusual, but fascinating environment, Movile Cave is the first known chemosynthesis-based groundwater ecosystem. Therefore, Movile Cave deserves stringent attention and protection.

**Keywords:** Movile Cave; Romania; subterranean biodiversity; chemoautotrophically based; groundwater ecosystem





## 1. Introduction

Movile Cave is an underground void, which has no natural opening to the surface. It is located on the outskirts of the town of Mangalia (SE Romania), 2 km from the Black Sea shore [1] and it was intercepted 18 m below the surface by an artificial geological survey shaft dug in June 1986. The cave is developed in Sarmatian limestones (12.5 MY), which were covered by Quaternary deposits (clays and loess), approximately 2.5 million years ago [2], and it was formed by classical karst dissolution processes combined with sulfuric acid speleogenesis (SAS), a process mediated by sulfur-oxidizing bacteria [3]. Additional geographical description information and geological data is presented in Sarbu et al. 2019 [1]. The cave appears as a horizontal maze, with a total development of 240 m, and consists of two levels (Figure 1). The upper level is dry and it lacks speleothems. The atmosphere in the upper level is warm (21 °C) and it contains 19% dioxygen ($O_2$) and 1% carbon dioxide ($CO_2$). The lower level is 40 m long and partially flooded, with the Lake Room and three Air-Bells as the only aerated zones. The $O_2$ concentration decreases gradually in the Air-Bells to 7%, while the $CO_2$ concentration increases to 3.5% due to the activity of microorganisms, which form biofilms [4,5]. In the Air-Bells, the $O_2$ dissolved in water originates from the cave atmosphere and it is rapidly used for the oxidation of hydrogen sulfide ($H_2S$) and methane ($CH_4$). Below a depth of 1 mm, the water becomes

completely anoxic [6]. The water is relatively stagnant at the surface in the Lake Room and in the nearby Air-Bells, while some flow (i.e., $5\,l\,s^{-1}$) was detected at depths over 1 m in the flooded cave passages [3].

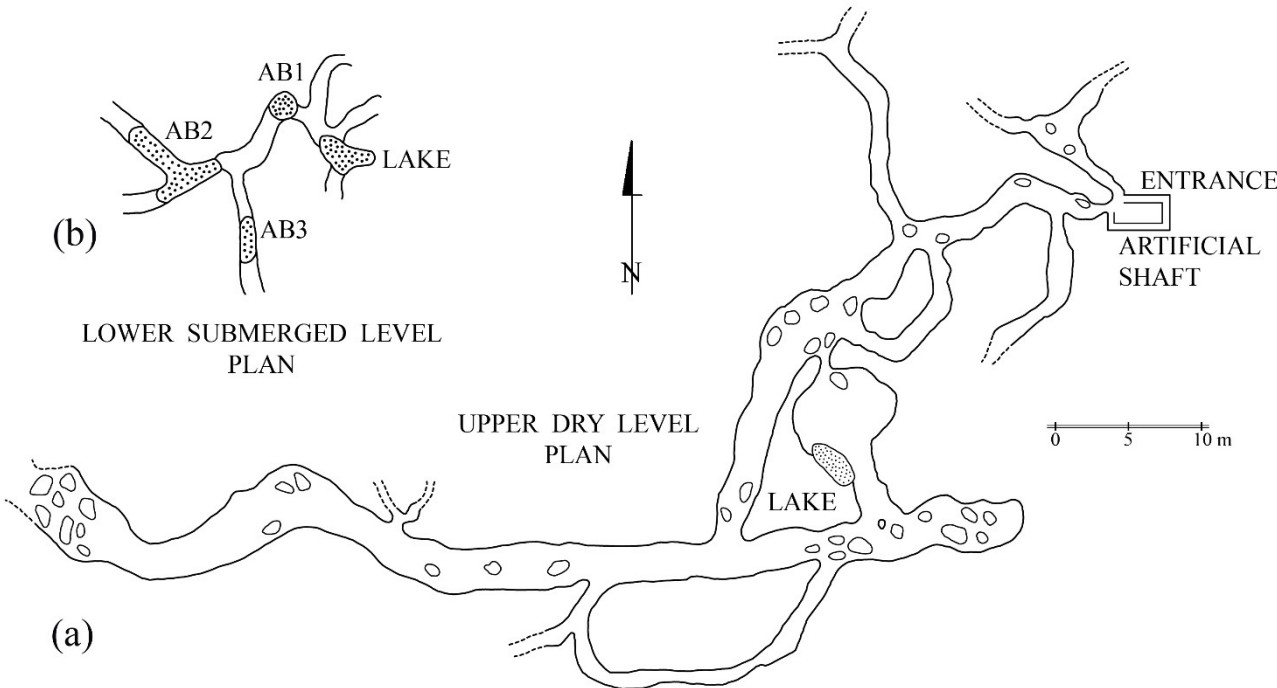

**Figure 1.** Plan of Movile Cave (**a**) with depiction of the water surfaces (dotted areas) in Lake Room and the three Air-Bells (AB) (**b**).

Movile Cave is the first subterranean groundwater ecosystem described to be solely based on chemosynthesis [7]. Reduced chemical compounds such as $H_2S$, $CH_4$ and ammonium ($NH_4^+$) are supplied continuously and in large amounts by the thermomineral water that ascends along geological faults from an artesian aquifer [5] located at a depth of 180 to 200 m in Mesozoic limestones [2]. Movile Cave is a window of access to a groundwater aquifer that occupies a surface area of approximately 50 to 100 $km^2$ and supports a particular hypogean ecosystem. Other windows of access to the aquifer are the old hand-dug wells in the town of Mangalia and in the surrounding villages, as well as several thermal sulfidic springs located along the sea-shore, at distances of 1 to 3 km from Movile Cave. Several of the endemic species that inhabit Movile Cave were also encountered in these springs and wells.

Along with $CH_4$ and $NH_4$, $H_2S$ represents the energy source for a wide variety of microorganisms [8] that use $O_2$, nitrate, sulfate, and ferric iron ($Fe^{3+}$) as electron acceptors [9–11]. The carbon fixed in situ by microorganisms, either swimming freely or congregated in thick biofilms, represents the food base of the trophic webs in Movile Cave. Aquatic invertebrates that graze on microorganisms, roam at the water surface where $O_2$ is available in small concentrations. Terrestrial invertebrates are rarely found in the upper dry sections of the cave, but they are unusually abundant in the lower, partially flooded cave level (Figure 1), in the Lake Room and the Air-Bells, where isopods, pseudoscorpions, millipedes, and insects feed on nearly any organic debris or graze on soil microorganisms, while predatory chilopods and spiders chase their prey, consisting of isopods, collembola, beetles or other spiders.

Analogous chemosynthesis-based cave ecosystems, such as those in the Frasassi caves (Italy) [12–14], Melissotrypa Cave (Greece) [15], Ayyalon Cave (Israel) [16,17], and Tashan Cave (Iran) [18–20], display comparable diversities of cave dwelling invertebrates with large numbers of endemisms (Table 1). These ecosystems also depend primarily on in

situ carbon fixation by chemoautotrophic microorganisms using the H$_2$S present in water. The natural resources in these ecosystems sustain, just like in Movile Cave, the growth of diverse and complex microbial communities, that form biofilms.

**Table 1.** The number of invertebrate species encountered in sulfidic cave ecosystems analogous to Movile Cave. All endemic species encountered in these sulfidic ecosystems are restricted to caves.

| Cave | Species Present | Endemic Species |
|---|---|---|
| Movile Cave (Romania) | 52 | 37 |
| Frasassi caves (Italy) | 56 | 16 |
| Melissotrypa Cave (Greece) | 30 | 8 |
| Ayyalon Cave (Israel) | 8 | 7 |
| Tashan Cave (Iran) | 3 | 3 |

These ecosystems can be appropriate examples for the ecological theory regarding the diversity-driven speciation [21] and convergent or parallel evolution. On the contrary, in caves with energy input from the surface, thus scarcer and less diverse food resources, the diversity of cave organisms is significantly lower.

The purpose of this study is to provide an up-to-date list of invertebrate cave-dwelling species living in the peculiar Movile Cave ecosystem and to draw attention to the scientific importance of these species and their fascinating habitat.

## 2. Movile Cave Fauna

### 2.1. Aquatic Fauna

In Movile Cave, the food base for aquatic invertebrates is produced autochthonously, and consists of microorganisms that thrive in the sulfide-rich water and sediments and use the energy resulting from the oxidation of the reduced chemical compounds from the thermo-mineral water [8].

The microorganisms swim freely in water as bacterioplankton, or they gather in thick biofilms floating at the water surface or they attach to rock surfaces (Figure 2). These represent a copious food source for the numerous consumers thriving in this groundwater ecosystem [7]. Various types of Archaea [10], sulfur-oxidizing bacteria [8], methanotrophs [9,10,22–24], or nitrifying and denitrifying bacteria [10], have been identified in Movile Cave. Representatives of a newly described strain of *Thiovulum* swim actively at the water surface and gather in loose veils [25]. Epi-symbiotic strains of *Thiothrix* sp. live on the bodies of aquatic amphipod crustaceans [11]. A diverse fungal community, associated with microbial mats and submerged sediments, is also present [26]. Underneath the floating mats formed by microorganisms, a diverse community of aquatic invertebrates feast on the abundant and assorted food provided [1,7].

Flagellate, ciliate, and amoebozoan protists and rotifers are the smallest grazers in this peculiar groundwater ecosystem [27]. They feed on bacterioplankton and microbial biofilms. Microorganisms, either pro- or eukaryotic, are consumed by the abundant meiofauna consisting of rotifers, nematodes, polychaetes, and copepod and ostracod (Figure 3C) crustaceans. Among the aquatic invertebrates, the nematodes (*Panagrolaimus* sp. and *Poikilolaimus* sp.) are important prey for cyclopoid copepods *Eucyclops greateri scythicus* [28]. Groups of hundreds of Moitessierid gastropods *Heleobia dobrogica* (Figure 3E) gather at the edge of the water, along the lake banks feeding on microbial biofilms [29]. Crustaceans of the Ostracoda, Copepoda, Amphipoda, and Isopoda (Figure 3B) swim at the water surface or slink on the submerged cave walls, and feed on smaller organisms they come across (Table 2). Eyeless and unpigmented water scorpions (*Nepa anophthalma*, Hemiptera), one of the top predators in this aquatic environment (Figure 3D), hide cautiously under the water surface, between the lake walls asperities, and wait for prey consisting of amphipods and isopods. Leeches (*Haemopis caeca*) are also considered top predators in this aquatic ecosystem (Figure 3A). They swim elegantly in the mass of water, diving deeper sometimes, and approach the lake banks where they predate on earthworms (*Helodrilus* sp.) that live in

high numbers in the sediments. Flat worms (*Dendrocoelum obstinatum*) glide on sediments in shallow water near the lake banks, never deeper than 1 cm, where some dissolved $O_2$ is still present, or swim at the water surface [30]. They graze on microorganisms, or they can predate on worms and crustaceans [1].

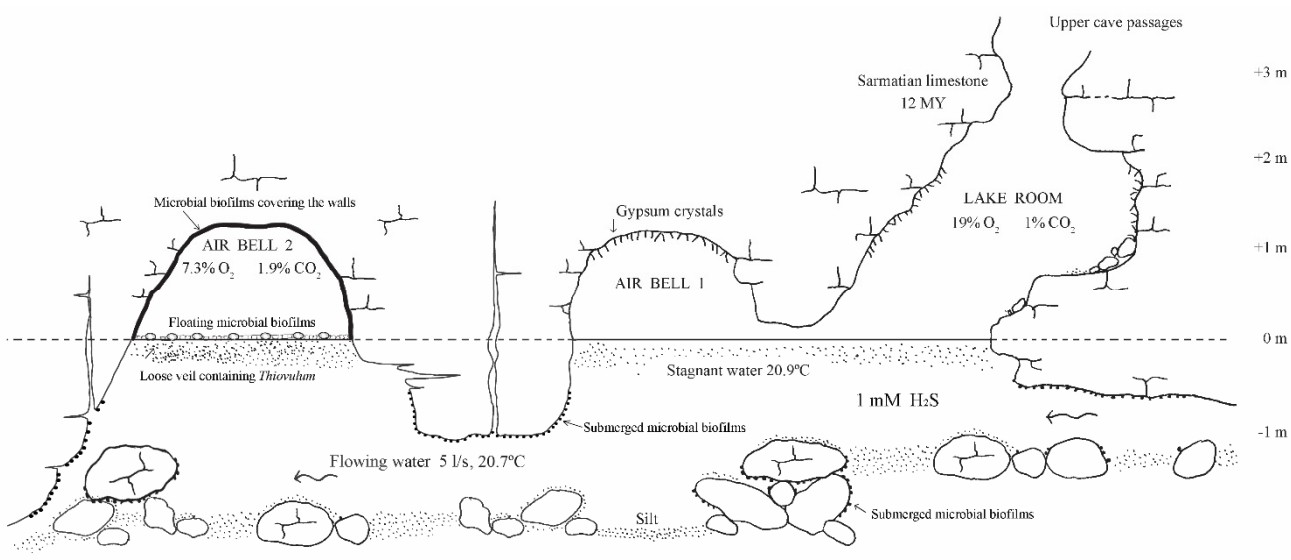

**Figure 2.** Longitudinal profile through the submerged cave gallery (modified after Sarbu and Popa, 1992). Some water flow is present at depths greater than 1 m, while at the surface, the water is rather stagnant and supports the growth of microorganisms in thick or loose biofilms. Anaerobic microbes attach to the cave walls, rocks, and sediments.

The trophic chains in Movile Cave are likely simple compared to the complex food webs in aboveground ecosystems with more complex interactions. The in-situ food production, the diversity of microorganisms and invertebrates in Movile Cave, are remarkably rich [1] for a cave environment. Conversely, the majority of caves resulting from epigenetic karstic dissolution processes, where the base of the food web is represented by input of allochthonous food of photoautotrophic origin, host lower numbers of cave-adapted species. Using stable isotope analysis, Sarbu et al. [5,7] provided a first diagram of the aquatic food web of Movile Cave. Its base is represented by the microbial biofilms formed mainly as a result of $H_2S$ and $CH_4$ oxidation. The primary consumers are terrestrial grazers (*Archiboreoiulus serbansarbui*, *Trachelipus troglobius* and *Armadillidium tabacarui*), and aquatic grazers (*Helodrilus* sp., *Niphargus racovitzai*, *Niphargus dancaui*). These are predated upon by the secondary consumers and top predators (aquatic) *Nepa anophthalma* and *Haemopis caeca*; (terrestrial) *Medon dobrogicus*, *Agraecina cristiani* and *Cryptops speleorex*.

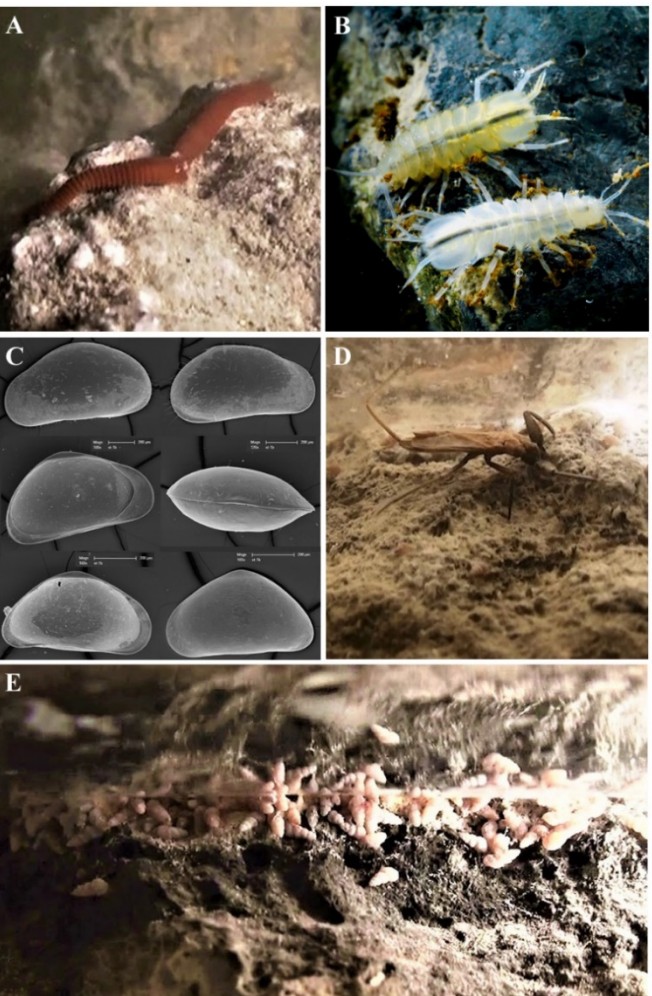

**Figure 3.** Aquatic invertebrates present in Movile Cave; (**A**). the leech (*Haemopis caeca*); (**B**). isopod crustaceans (*Asellus aquaticus infernus*); (**C**). ostracods (*Pseudocandona* sp. nov.), the scale length is 200 μm; (**D**). water scorpions (*Nepa anophthalma*); (**E**). gastropods (*Heleobia dobrogica*).

### 2.2. Terrestrial Fauna

The terrestrial fauna is more complex. It is composed of four species of isopods, six spider species, four pseudoscorpions, one acarian species, three chilopods, two millipedes, three springtails, two dipluran species, and five beetles (Tables 2 and 3). The largest invertebrate species and top predator in Movile Cave ecosystem is *Cryptops speleorex* (Figure 4B) [31]. These voracious centipedes are 8-10 cm long, and they roam continuously in search for prey, which is not scarce, and it ranges from the smallest collembolan or coleoptera species to the stout isopods *Trachelipus troglobius* (Figure 4D). The geophilid centipedes *Geophilus* sp. and *Clinopodes carinthiacus* are also among the predators in this ecosystem. They chase smaller prey, such as collembola, smaller isopods or the offspring of the larger isopods, and pseudoscorpions.

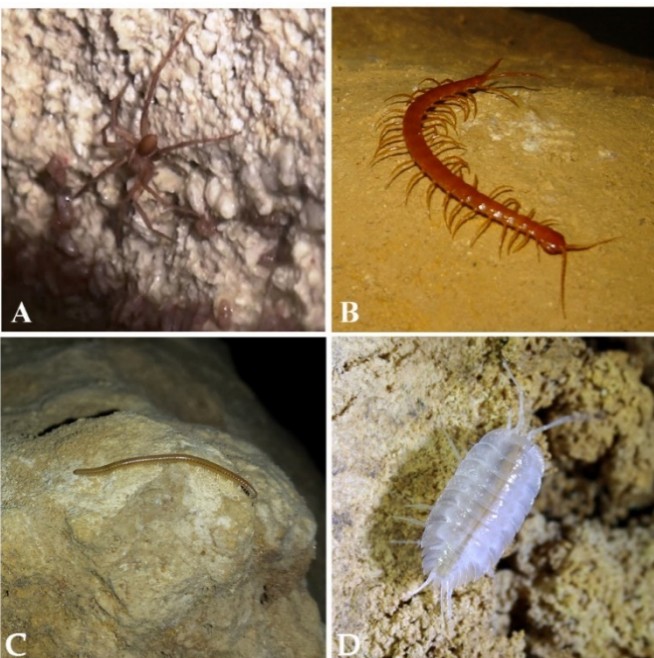

**Figure 4.** Terrestrial invertebrates present in Movile Cave; (**A**). the spider (*Agraecina cristiani*); (**B**). top predator chilopod (*Cryptops speleorex*); (**C**). millipede (*Archiboreoiulus serbansarbui*); (**D**). isopod (*Trachelipus troglobius*).

Numerous collembola (springtails) are present on the water surface, on top of the floating microbial biofilms, as well as on the biofilms that cover the cave walls in the remote Air-Bells [32,33]. These are very small hexapods, and therefore, they do not count much in the total biomass available for higher trophic levels. Instead, the three collembola species are significant for very high numbers. They achieve large densities because of the availability of luxurious resources, both in terms of type and amount of food. Collembola are everywhere in populated places in Movile Cave, such as in Lake Room or Air-Bells. Two of the three species jump continuously in all directions; therefore, they can easily become part of the menu of other cave inhabitants. Many of the small-size predators rely heavily on springtails as their primary food source.

The largest terrestrial species diversity and density in Movile Cave is present in Lake Room (Figure 2). Here, one can observe slender millipedes (*Archiboreoiulus serbansarbui*, in Figure 4C, and *Strongylosoma jaqueti*), the tiny *Haplophthalmus movilae* or the large and hunchbacked isopods *Trachelipus troglobius* (Figure 4D) approaching the lake banks for drinking, or more likely for grazing on microorganisms present along the lake shore, with the risk of being preyed upon by water scorpions (*Nepa anophthalma*). The millipedes and isopods are mostly present in the Lake Room and Air-Bell 1, most likely because $O_2$ is still abundant in these chambers. On the contrary, other isopods (i.e., *Caucasonethes vandeli pygmaeus* and *Armadillidium tabacarui*) are by far more abundant in the remote Air-Bells (2 and 3).

**Table 2.** List of troglobionts and stygobionts from Movile Cave.

| | Aquatic/Terrestrial | Species | Taxonomic Affiliation | References |
|---|---|---|---|---|
| 1 | Aquatic | *Dendrocoelum obstinatum* *; Stocchino et al., 2017 | Platyhelminthes, Dendrocoelidae | [30] |
| 2 | Aquatic | *Panagrolaimus* cf. *thienemani* * | Nematoda, Panagrolaimidae | [34] |
| 3 | Aquatic | *Chronogaster troglodytes* *; Poinar and Sarbu, 1994 | Nematoda, Chronogasteridae | [35] |
| 4 | Aquatic | *Haemopis caeca* *,#; Manoleli et al., 1998 | Annelida, Hirudinea, Haemopidae | [36] |
| 5 | Aquatic | *Helodrilus* sp. nov. * | Annelida, Clitellata, Lumbricidae | Martin, P., pers. comm. |
| 6 | Aquatic | *Heleobia dobrogica* *; Grossu and Negrea, 1989 | Gastropoda, Moitessieriidae | [29] |
| 7 | Aquatic | *Pseudocandona* sp. nov. * | Crustacea, Ostracoda, Cyprididae | Danielopol, D., pers. comm. |
| 8 | Aquatic | *Eucyclops graeteri scythicus* *; Plesa, 1989 | Crustacea, Copepoda, Cyclopidae | [37] |
| 9 | Aquatic | *Parapseudoleptomesochra italica*; Pesce and Petkovski, 1980 | Crustacea, Copepoda, Harpacticoida | Rouch, pers. comm. |
| 10 | Aquatic | *Niphargus racovitzai* *; Dancau, 1970 | Crustacea, Amphipoda, Niphargidae | [38] |
| 11 | Aquatic | *Niphargus dancaui* *,#; Brad et al., 2015 | Crustacea, Amphipoda, Niphargidae | [39] |
| 12 | Aquatic | *Asellus aquaticus infernus* *,#; Turk-Prevorčnik and Blejec, 1998 | Crustacea, Isopoda, Asellidae | [40] |
| 13 | Terrestrial | *Caucasonethes vandeli pygmaeus* *; Giurginca, 2020 | Crustacea, Isopoda, Trichoniscidae | [41] |
| 14 | Terrestrial | *Haplophthalmus movilae* *; Gruia and Giurginca, 1998 | Crustacea, Isopoda, Trichoniscidae | [42] |
| 15 | Terrestrial | *Trachelipus troglobius* *; Tabacaru and Boghean, 1989 | Crustacea, Isopoda, Trachelipodidae | [43] |
| 16 | Terrestrial | *Armadillidium tabacarui* *; Gruia et al., 1994 | Crustacea, Isopoda, Armadillidiidae | [44] |
| 17 | Terrestrial | *Chthonius monicae* *; Boghean, 1989 | Arachnida, Pseudoscorpiones, Chthoniidae | [45] |
| 18 | Terrestrial | *Chthonius borissketi* *; Curčić et al., 2014 | Arachnida, Pseudoscorpiones, Chthoniidae | [46] |
| 19 | Terrestrial | *Roncus dragobete* *; Curčić et al., 1993 | Arachnida, Pseudoscorpiones, Neobisiidae | [47] |
| 20 | Terrestrial | *Roncus ciobanmos* *; Curčić et al., 1993 | Arachnida, Pseudoscorpiones, Neobisiidae | [47] |

**Table 2.** *Cont.*

| | Aquatic/Terrestrial | Species | Taxonomic Affiliation | References |
|---|---|---|---|---|
| 21 | Terrestrial | *Palliduphantes constantinescui* *; Georgescu, 1989 | Arachnida, Araneae, Linyphiidae | [48] |
| 22 | Terrestrial | *Agraecina cristiani* *,#; Georgescu, 1989 | Arachnida, Araneae, Liocranidae | [48] |
| 23 | Terrestrial | *Kryptonesticus georgescuae* *; Nae, Sarbu, and Weiss, 2018 | Arachnida, Araneae, Nesticidae | [49] |
| 24 | Terrestrial | *Hahnia caeca* *; Georgescu and Sarbu, 1992 | Arachnida, Araneae, Hahniidae | [50] |
| 25 | Terrestrial | *Labidostomma motasi* *; Iavorschi, 1992 | Arachnida, Acarina, Labidostommatidae | [51] |
| 26 | Terrestrial | *Geophilus* sp. nov. * | Chilopoda, Geophilidae | Baba, St., pers. comm. |
| 27 | Terrestrial | *Cryptops speleorex* *,#; Vahtera et al., 2020 | Chilopoda, Cryptopidae | [31] |
| 28 | Terrestrial | *Archiboreoiulus serbansarbui* *,#; Giurginca et al., 2020 | Diplopoda, Julida, Julidae | [52] |
| 29 | Terrestrial | *Onychiurus movilae* *; Gruia, 1989 | Collembola, Onychiuridae | [53] |
| 30 | Terrestrial | *Oncopodura vioreli* *; Gruia, 1989 | Collembola, Oncopoduridae | [53] |
| 31 | Terrestrial | *Plusiocampa isterina* *; Condé, 1993 | Diplura, Campodeidae | [54] |
| 32 | Terrestrial | *Plusiocampa euxina* *; Condé, 1996 | Diplura, Campodeidae | [55] |
| 33 | Terrestrial | *Medon dobrogicus* *; Decu and Georgescu, 1994 | Coleoptera, Staphylinidae | [56] |
| 34 | Terrestrial | *Tychobythinus sulphydricus* *; Poggi and Sarbu, 2013 | Coleoptera, Staphylinidae | [57] |
| 35 | Terrestrial | *Decumarellus sarbui* *; Poggi, 1994 | Coleoptera, Staphylinidae | [58] |
| 36 | Terrestrial | *Bryaxis dolosus* *; Poggi and Sarbu, 2013 | Coleoptera, Staphylinidae | [57] |
| 37 | Terrestrial | *Clivina subterranea* *; Decu et al., 1994 | Coleoptera, Clivinidae | [59] |
| 38 | Aquatic | *Nepa anophthalma* *; Dedu et al., 1994 | Hemiptera, Nepidae | [60] |

*—species endemic to Movile Cave; #—species found in nearby springs and wells.

**Table 3.** List of troglophiles and stygophiles from Movile Cave.

| | Aquatic/Terrestrial | Species | Taxonomic Affiliation | References |
|---|---|---|---|---|
| 1 | Aquatic | *Udonchus tenuicaudatus*; Cobb, 1913 | Nematoda, Rhabdolaimidae | [34] |
| 2 | Aquatic | *Poikilolaimus* sp. | Nematoda, Rhabditidae | [34] |
| 3 | Aquatic | *Monhystrella* sp. | Nematoda, Monhysteridae | [34] |
| 4 | Aquatic | *Habrotrocha rosa*; Donner, 1949 | Rotatoria, Habrotrochidae | Ricci, C., pers. comm. |
| 5 | Aquatic | *Habrotrocha bidens*; Gosse, 1851 | Rotatoria, Habrotrochidae | Ricci, C., pers. comm. |
| 6 | Aquatic | *Aelosoma hyalinum*; Bunke, 1967 | Annelida, Aeolosomatidae | Dumnicka, E., pers. comm. |
| 7 | Aquatic | *Aelosoma italica*; Bunke, 1967 | Annelida, Aeolosomatidae | Dumnicka, E., pers. comm. |
| 8 | Aquatic | *Tropocyclops prasinus*; Fischer, 1860 | Crustacea, Copepoda, Cyclopidae | [37] |
| 9 | Terrestrial | *Carniella brignolii*; Thaler and Steinberger, 1988 | Arachnida, Araneae, Theridiiae | [61] |
| 10 | Terrestrial | *Dysdera hungarica*; Kulczynski, 1897 | Arachnida, Araneae, Dysderidae | Weiss, L., pers. comm. |
| 11 | Terrestrial | *Clinopodes carinthiacus*; Latzel, 1880 | Chilopoda, Geophilidae | Zapparoli, M., pers. comm. |
| 12 | Terrestrial | *Strongylosoma jaqueti*; Verhoeff, 1898 | Diplopoda, Paradoxosomatidae | Tajovsky K., pers comm. |
| 13 | Terrestrial | *Pygmarrhopalites pygmaeus*; Wankel, 1860 | Collembola, Arrhopalitidae | [62] |

*Armadillidium tabacarui* form here large populations of up to 200 individuals per square meter. The density of *Caucasonethes vandeli pygmaeus* is practically impossible to estimate in Air-Bells 2 and 3 especially as the researcher must continue to breathe through a SCUBA regulator, and to wear a diving mask. *Caucasonethes vandeli pygmaeus* is an extremely small isopod, less than 2 mm long, it is translucid, and moves very fast.

Larger spider species, such as *Agraecina cristiani* (Figure 4A) and *Dysdera hungarica*, predate mainly on collembola, but also on smaller isopods (*Caucasonethes vandeli pygmaeus*, *Haplophthalmus movilae*, and *Armadillidium tabacarui*), which are present on the cave floor and on the walls in relatively high numbers, along with the smaller spiders and pseudoscorpions. *Palliduphantes constantinescui* and *Kryptonesticus georgescuae* are web-weaving spiders that catch small prey such as collembola or small isopods. Even smaller spiders (*Hahnia caeca* and *Carniella brignolii*) are content with springtails and mites.

Five Coleoptera species are present in Movile Cave. Of these, *Medon dobrogicus* and *Clivina subterranea* are frequently present in Air-Bells 2 and 3 where they form large populations. They run continuously on the walls in these sections of the cave, and predate upon collembola, juvenile individuals of the isopods *Armadillidium tabacarui*, *Caucasonethes vandeli pygmaeus*, and *Haplophthalmus movilae*, or they can even chase the small *Chthonius monicae* pseudoscorpions. Due to its very small size, the latter can only feed on springtails and mites, or the small *Caucasonethes vandeli pygmaeus* isopods. Smaller, but also predatory staphylinids beetles (*Tychobythinus sulphydricus*, *Decumarellus sarbui*, and *Bryaxis dolosus*), can only hunt and feed on collembola.

Movile Cave is a unique habitat that needs unquestionable protection. Pollution of the groundwater aquifers by the intensive farming in South-Eastern Dobrogea, or spilling of various contaminants in the environment as a result of the expansion of the residential areas around Mangalia, can lead to severe disturbances of this exceptional ecosystem. The cave is already part of Natura 2000 sites (Code ROSCI0114), it is accessible only for scientific research, and it can only be entered based on authorized permissions. New technological advances in research methods allow for better understanding of how life can prosper even in such extreme environments, like Movile Cave, in total darkness, low pH, hypoxia and anoxia, high sulfide-, $CH_4$, and $CO_2$ concentrations. Microbiological research has evolved significantly from characterization of enzymes produced by microbes and cultivation of sulfide oxidizers [63], to the first molecular characterization of microbial communities by basic fingerprinting techniques and generation of clone libraries [10,22], to the nowadays Next-Generation Sequencing approaches that allow the examination of tens of thousands of sequences or complete genomes [25,64]. Regarding the invertebrates, new and undescribed species are no longer a great surprise, but they have to be identified and studied before their possible disappearance. It also allows to document the extent and nature of evolutionary convergence across distinct lineages of stygobiontic crustaceans and to determine to what extent natural selection was the driver of the extreme modifications observed in certain species thriving in the cave. The structure of the food web in this special environment is being studied by direct observations of the feeding behaviors, stable isotope analysis, and metagenomic investigations of the gut content of the species that inhabit the cave. Information on food webs is important and tells ultimately on how organisms can find ways to survive, and even to thrive, in ecosystems that do not depend on Sun-derived energy. Here, the food is produced in situ by using natural and inexhaustible energy sources, such as the $H_2S$, $CH_4$ and $NH_4^+$ from the deep subterranean aquifer. Finally, research on symbiosis between crustaceans and bacteria with biotechnological applications, discovery of new species of cave bacteria with possible antibiotic resistance, and experiments on their possible use for human health, has also a great potential to be explored in more detail in the future. Therefore, Movile Cave still has a lot more to offer.

**Author Contributions:** All three coauthors (T.B., S.I. and S.M.S.) have equal contribution in the conceptualization, data curation, writing and reviewing of this manuscript. All authors have read and agreed to the published version of the manuscript.

**Funding:** T. Brad was supported by a grant of the Ministry of Research and Innovation in Romania, project number PN-III-P4-ID-PCCF-2016-0016 (DARKFOOD), and by EEA Grants 2014-2021, under Project contract no. 4/2019 (GROUNDWATERISK). S. Iepure and S. Sarbu were supported by grants of Ministry of Research and Innovation (UEFISCDI) projects number PN-III-P4-ID-PCE-2020-2843 (EVO-DEVO-CAVE) and PN-III-P4-ID-PCCF-2016-0016 (DARKFOOD).

**Institutional Review Board Statement:** Not applicable.

**Informed Consent Statement:** Not applicable.

**Data Availability Statement:** No new data were created or analyzed in this study. Data sharing is not applicable to this article.

**Acknowledgments:** The authors acknowledge the support of the Group for Underwater and Speleological Exploration (GESS) for logistics, access to the cave and laboratory house in Mangalia, where preliminary processing of samples occurred. We are grateful to our collaborators which have identified the various species present in Table 2 (personal communications).

**Conflicts of Interest:** The authors declare no conflict of interest.

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
