# Peer review of "The Chemoautotrophically Based Movile Cave Groundwater Ecosystem, a Hotspot of Subterranean Biodiversity"

_diversity, doi:10.3390/d13030128_

Round 1

Reviewer 1 Report

This paper presents a description of a very unusual cave community and provides an updated faunal list.  Much of the material has been previously presented by the authors in other publications.  The authors could provide numerous compelling reasons why Movile Cave is worth protection beyond its date of discovery. I would like to have learned more about future efforts to e=increase our understanding of its bizarre ecosystem.  However, it remains a fascinating cave and it is clear from this paper that discoveries continue.  I have attached a document with brief comments and edits.

Author Response

Dear reviewer. Thank you for reading our manuscript. We hereby resubmit our work with your comments addressed and your suggestions included in the manuscript. More complex and elaborated maps of Movile Cave are available, but for our manuscript that focuses mainly on the diversity of the cave's biological community, the authors believe that figures 1 and 2 (the plan of the cave and a profile through Lake Room and Air-Bell 2), may be sufficient, as all invertebrate species are billeted here, in Lake Room and Air-Bells 1, 2 and 3.  There is a fourth Air-Bell that was visited once only, due to its very difficult accessibility, and no biological observations were performed there. We chose to not superimpose the maps of the upper and lower levels as this would result in an overly busy, hard to read map. The water level in Lake Room and Air-Bells is at a quote of 1.5 meters above sea level, and this groundwater aquifer drains slowly towards the shore of the Black Sea. Historically, the piezometric level of the Southern Dobrogean sulfidic groundwater aquifer has always been influenced by the level of the Black Sea.

Reviewer 2 Report

Congratulations to the authors for a well done review article on interesting organisms in a specific environment.

In 132 lines you write about "blind" water scorpions. I suggest the term “eyeless” scorpion.

Author Response

Thank you for reading our work and for you kind words. Your suggestions are now included in the manuscript.

Reviewer 3 Report

The manuscript submitted by Brad et al. nicely presents a review on the work performed so far on the biodiversity of such a special place, i.e., the Movile cave.

Several minor comments:

  • the authors should discuss briefly how were the papers chosen to be included in the review (e.g., what databases were consulted, if there were papers excluded and why, etc.);
  • figures 1 and 2 should have a better resolution; they are difficult to read;
  • L114: replace "invertebrate's" with "invertebrates";
  • L148: please check the style of the journal, if "in situ" needs to be set in italics.

Author Response

Thank you for reading our manuscript. We hereby resubmit our work with your comments addressed and your suggestions included in the manuscript. All articles regarding Movile Cave fauna were considered and cited. No paper on Movile Cave fauna was excluded.